# Aortic Valve Replacement: Understanding Predictors for the Optimal Ministernotomy Approach

**DOI:** 10.3390/jcm12216717

**Published:** 2023-10-24

**Authors:** Francesco Giosuè Irace, Ilaria Chirichilli, Marco Russo, Federico Ranocchi, Marcello Bergonzini, Antonio Lio, Francesca Nicolò, Francesco Musumeci

**Affiliations:** Department of Cardiac Surgery and Heart Transplantation, San Camillo Forlanini Hospital, Viale Gianicolense 87, 00151 Rome, Italyantoniolio@hotmail.it (A.L.); fr.musumeci@gmail.com (F.M.)

**Keywords:** aortic valve, aortic valve replacement, minimally invasive cardiac surgery, computed tomography

## Abstract

Introduction. The most common minimally invasive approach for aortic valve replacement (AVR) is the partial upper mini-sternotomy. The aim of this study is to understand which preoperative computed tomography (CT) features are predictive of longer operations in terms of cardio-pulmonary bypass timesand cross-clamp times. Methods. From 2011 to 2022, we retrospectively selected 246 patients which underwent isolated AVR and had a preoperative ECG-gated CT scan. On these patients, we analysed the baseline anthropometric characteristics and the following CT scan parameters: aortic annular dimensions, valve calcium score, ascending aorta length, ascending aorta inclination and aorta–sternum distance. Results. We identified augmented body surface area (>1.9 m^2^), augmented annular diameter (>23 mm), high calcium score (>2500 Agatson score) and increased aorta–sternum distance (>30 mm) as independent predictors of elongated operation times (more than two-fold). Conclusions. Identifying the preoperative predictive factors of longer operations can help surgeons select cases suitable for minimally invasive approaches, especially in a teaching context.

## 1. Background

Surgical aortic valve replacement (SAVR) historically represents the treatment of choice for patient affected by severe aortic valve disease. A careful patient evaluation and the accurate establishment of operative risk according to age, general conditions, co-pathologies and anatomy drive the heartteam-based decision over to perform surgical aortic valve replacement (SAVR) or transcatheter aortic valve replacement (TAVR) [1,2,3,4]. Current European guidelines suggest performing SAVR in patients younger than 75 years old, who have low surgical risk according to EuroSCORE II (<4%) and STS-SCORE (<4%) calculations. Patients older than 75 years old or intermediate–high surgical candidates may undergo TAVR after a careful patient evaluation, anatomy assessment and heart team discussion. A tailored approach should also be taken into consideration in all borderline cases according to the centre’s expertise and the patient’s wish [5].

As technology and interventional techniques rapidly evolved during the last decade, with the miniaturization of valve prostheses and delivery systems, surgery as well has seen an increase in the use of minimally invasive techniques to treat valve disease. Accordingly, operative outcomes and results have been largely and progressively improving. A minimally invasive aortic valve replacement may be performed with several skin incisions like upper mini-sternotomy, right anterior mini-thoracotomy or axillary mini-thoracotomy. Recently, Wilbring et al. reported a propensity-matched study on minimally invasive SAVR, describing thecomparable safety of median sternotomy and the association with a faster discharge time from ICU to home, shorter ventilation times and less blood loss which requirestransfusions [6].

Nevertheless, recently, robotic application in the field of aortic valve disease has been pioneered [7]. The role of minimally invasive techniques on patient outcomes have been largely reported; however, no specific recommendations are present in the guidelines and no randomized data have been reported to compare M-SAVR with TAVR [8].

As we learned from the “TAVR Experience”, a careful preoperative evaluation of a patient’s anatomic characteristics with a deep dive into CT scan assessment is crucial to properly select the valve to implant, to perform the procedure and to optimize patient’s outcome. A 3DCT analysis with the dedicated software (3mensio Structural Heart v10.0, Pie Medical Imaging, Maastrict, The Netherlands) represents the basis for the work-up of a TAVI patient.

A systematic CT anatomic assessment is less used in open surgery and all the surgical steps are usually and historically guided from the surgeon’s finding and expertise inthe operative field. We are supposed to screen all patients undergoing aortic valve surgery with 3mensio analysis in order to preoperatively obtain anatomic details like valve morphology, annular dimension, annular perimeter, calcium score, aortic angle and working distance from the sternum, which may have a role in the procedural planning and in the procedural steps. Since the Achilles’heelof all minimally invasive cardiac surgeries is represented by the elongation of surgical and CPB times, we aimed to connect a patient’s characteristics with operative features and outcomes.

Traditional surgical aortic valve replacement through a median full sternotomy has proven to be a safe and effective procedure to treat patients with severe aortic valve pathology, with excellent long-term results [9]. However, in the last two decades, minimally invasive aortic valve replacement has become an increasingly performed approach as an alternative to conventional full sternotomy due to its significant advantages in terms of reduced mortality, morbidity and shorter hospital stay [10,11]. However, minimally invasive aortic valve surgery is usually associated with a limited exposure which could result in some technical challenges. For this reason, a careful preoperative evaluation of a patient’s anatomic characteristics with a standardized CT scan assessment is crucial in the proper patient selection process in order to perform the most appropriate procedure and optimize a patient’s outcomes. As has already been demonstrated, the preoperative CT scan plays a central role in improving the procedural success rate and reducing the incidence of sternotomy conversion [12]. The traditional surgical approach is usually guided by the surgeon’s expertise, and a systematic preoperative CT scan evaluation is not mandatory. On the contrary, in minimally invasive aortic valve surgery, the evaluation of some parameters, such as aortic annular dimensions, aortic valve calcium score, ascending aorta length, aortic annular inclination and the sternum/STJ distance, may have an important role during the planning phase.

The aim of the present study is to assess a patient’s anatomic characteristics using a TAVR-like CT scan protocol to determine surgical candidacy for a minimally invasive aortic valve replacement and to define the risk factors for complex surgical procedures. The final scope is to optimize the patient selection criteria and improve surgical outcomes.

## 2. Patients and Methods

We retrospectively analysed all patients who underwent isolated aortic valve replacement through upper mini-sternotomy at San Camillo Forlanini Hospital from January 2011 to December 2021. Both aortic valve stenosis and insufficiency were included. A total of 440 patients were screened: reoperations, patients who implanted rapid-deployment prosthesis and patients who needed aortic annulus enlargement were excluded, as well as patients without a preoperative ECG-gated CT scan. A sample of 246 patients were included in the study (Figure 1).

Electronic records for each patient were assessed for anthropometric features, preoperative aortic valve characteristics and intraoperative data: type of prosthesis, cannulation strategy, cardiopulmonary bypass (CPB) and cross-clamp (X-clamp) times.

All included patients underwent preoperative ECG-gated CT; most of the patients treated at our institution receive a cardiac CT scan in the preoperative work up, both for coronary study, avoiding the need of coronary angiography when possible, and for anatomic assessment (aortic position, aortic wall calcification). The CT protocol included an ECG-retrospective contrast-enhanced CT scan of the entire thoracic aorta after injection of a bolus of 70–90 mL of iomeprol 400 mg L/dL (Iomeron 400, Bracco, Milan, Italy) at a flow rate of 5 mL/s, followed by a 30 mL saline flush, using a 256-slice or 64-slice CT scanner. CT data sets were reconstructed with a slice thickness of 1 mm (reconstruction increment 0.5 mm) at 70% of the RR interval using a medium-smooth convolution algorithm and an image matrix of 512 × 512 pixels. 

The studies were re-analysed, and the images were reconstructed with 3mensio software (v10.0, Pie Medical Imaging, Maastricht, The Netherlands). The following parameters were systematically assessed: aortic annulus dimensions (obtained in the double oblique plane after multi-planar reconstruction),aortic valve calcium score (calculated by the software), ascending aorta length from aortic annulus to the origin of the brachiocephalic trunk (measured on the reconstructed stretched vessel), inclination angle of the aortic annulus (considered to be the angle between the aortic annulus plane and the horizontal plane) and the distance between the sternum (posterior ridge) and the aorta at the level of the sino-tubular junction (Figure 2).

### 2.1. Surgical Technique

All selected patients were operated on through an upper J-ministernotomy at the 4th intercostal space (Figure 3). CPB was established either with central arterial (distal ascending aorta) and venous (right atrium/inferior vena cava) cannulation or peripheral cannulation (common femoral artery and vein) or with a combination of them. Antegrade haematic warm cardioplegia was used, and repeated every 15–20 min, or the del Nido solution was used, according to surgeon preference. The aorta is then opened at the level of the sino-tubular junction with a “hockey stick” incision towards the non-coronary sinus; the native valve is excised and the annulus is decalcified according to the anatomy. After conventional sizing, the prosthetic valve of choice is implanted with separated 2/0 Ticron-pledgeted sutures. Aorta is sutured with double layer 4/0 Prolene running sutures. Then, the operation is completed in standard fashion. All the operations were performed by experienced surgeons (>50 aortic valve replacements).

### 2.2. Statistical Analysis

All data processing was carried out on a workstation running IBMSPSS 26 (Armonk, NY, USA) on a Windows 10 machine (Microsoft Corp, Redmond, WA, USA). Categorical variables were presented as numbers and percentages. Continuous variables were expressed as mean ± standard deviation (SD). Normality of the data was assessed using the Shapiro–Wilk test. Univariate and multivariate logistic regression analyses were performed to identify outcome predictors. The cut-off choice was made by creating ROC curves based on the variables that were statistically significant in the regression analysis, and further estimation of an odds ratio was performed through crosstabulation. 

## 3. Results

### 3.1. Patients and Operative Characteristics

Patient baseline characteristics are summarized in Table 1. They were predominantly males (126, 51.2%); the mean age was 67 ± 12 years old. The mean body surface area (BSA) was 1.82 ± 0.19 m^2^ and the mean body mass index was 26.6 ± 3.8 kg/m^2^. A total of 198 patients (80.5%) had prevalent aortic stenosis and 62 (25.2%) patients had a bicuspid aortic valve (BAV). 

The mean size of prostheses implanted was 22.7 ± 2 mm; 223 (90.7%) patients received a biological prosthesis. The arterial cannulation strategy was central (ascending aorta) employed in 203 cases (82.5%) and peripheral (femoral artery) in 43 cases (17.5%); for venous cannulation, 127 patients (51.6%) had central (right atrium) and 119 (48.4%) had femoral vein cannulation.

The overall mean CPB time was 85.3 ± 20.3 min and the X-clamp time was 64 ± 15.9 min.

Three patients (1.2%) required conversion to full sternotomy after weaning from the CPB for major bleeding issues.

### 3.2. CT Scan Findings

The mean aortic annulus dimensions were 24 ± 2.8 mm in average diameter and 77.1 ± 8.9 mm in perimeter. The mean valve calcium score was 2700 ± 2260 Agatson units. The mean ascending aorta length, from the aortic annulus to the origin of the innominate artery was 89.1 ± 12 mm, while the mean distance between the sternum and the aorta was 33.7 ± 10 mm (Table 2). 

### 3.3. Predictors of Longer Procedures 

First, to consider a longer, and probably more difficult, procedure we identified cut-offs on CPB and X-clamp times: we chose to designate “longer procedure” for those procedures whose operating times are above the 75th percentile of *both* CPB and X-clamp times (Figure 4). Accordingly, 48 patients (19.5%) underwent a “*longer* procedure”.

Then, we ran a regression analysis and identified the following preoperative predictors of a longer procedure: male sex was associated with longer operative times (HR 2.484, 95% CI:1.270–4.859, *p*: 0.008); all the anthropometric characteristics (height, weight, BSA and BMI) were also positively associated with longer procedures, with BSA being the strongest predictor (HR 11.41, 95% CI: 2.063–63.11, *p*: 0.005).

For the CT scan-derived parameters, we found that annular dimensions were predictive of longer operations, with the average diameter as a stronger predictor (HR 1.164, 95% CI: 1.021–1.327, *p*: 0.023) as well as the calcium score (HR 1.696, 95% CI: 1.088–2.644, *p*: 0.020) and the aorta–sternum distance (HR 1.052, 95% CI: 1.014–1.092, *p*: 0.007). Calcium score values were analysed after logarithmic transformation. The presence of the bicuspid aortic valve was not associated with longer operative times (HR 0.495, 95% CI: 0.190–1.293, *p*: 0.151). 

All the regression analyses data are summarized in Table 3.

For the continuous variables identified as possible predictors, we used ROC curves to extrapolate optimal cut-off points. The relative diagrams are showed in Figure 5. We selected BSA as anthropometric variable and we selected the average annulus diameter for the aortic annular dimensions. After identification of the optimal cut-off points (BSA = 1.9 m^2^, annulus diameter = 23 mm, calcium score = 2500 Au, sternum/STJ distance = 30 mm), we extrapolated the relative odds ratio values for each variable: body surface area OR 3.2 (*p*: <0.01), average annular diameter OR 2.7 (*p*: <0.01), calcium score OR 2.9 (*p*: <0.01) and sternum-STJ distance OR 2.2 (*p*: 0.02).

Comprehensive values of independent predictors of longer operation times and relative odds ratios are summarized in Table 4; the same variables are also shown in Table 5 in the form of a comparison between the two groups of short (below the 75th percentile) and long (above the 75th percentile) procedures. 

## 4. Discussion

In our experience, we found several independent predictors of longer CPB and X-clamp times. In particular, among all anthropometric characteristics, BSA has proven to be the strongest predictor, with a cut-off point of 1.9 m^2^; among CT scan anatomic parameters, an aortic annular diameter > 23 mm, a calcium score > 2500 Agatson and anaorta–sternum distance > 30 mm were found to increase the operative times more than two-fold. These results are easy to explain; indeed, a very calcified aortic valve will take longer to be decalcified; as well, a higher annular diameter will require more annular stitches and consequently lead to longer times. Similarly, a huge aorta–sternum distance will limit the exposure and the operative accessibility due to a deeper site. Interestingly, bicuspid aortic valves are not associated with longer operative times, and this is partly in contrast with the “TAVR Experience” in which bicuspid aortic valves generally require a more accurate implantation. In 2018, Elattar et al., in their single-centre retrospective study, found that the access angle is strongly associated with procedure complexity; indeed, an access angle of 38° was a good cut-off to distinguish simple from complex minimally invasive aortic valve replacement [10]; furthermore, in their work, a different kind of angle was considered: Elattar et al. considered the angle between the aorta and the surgical access plane, while, in our cohort, we considered the inclination of the aortic annulus on the horizontal plane. On the contrary, Boti et al., in 2019, found that the access angle was not associated with procedure complexity [13].In other recent studies focused on the topic, Jug et al., in 2021, analysed similar anatomical features in the context of anterior mini-thoracotomy or upper mini-sternotomy [14], but they used only rapid-deployment valve prostheses. It is interesting that in their cohort they found that smaller aortic annulus was associated with longer operative times; this difference is probably due to a longer time spent in positioning the sutures with the standard technique in the larger annulus (as in our cohort). As welll, Charchyan and colleagues found how the aortic–sternum relationship and obesity was associated with more complex procedures [15]. In addition, in going beyond the simple preoperative assessment, scoring systems and computer based for approach planning were also developed following similar findings [16,17].

## 5. Limitations

Despite our research covering a 10-year time span, many patients could not be included due to its retrospective nature, underpowering the study. This results in lower strength of our data; in fact, the identified predictors were not significant when aggregated in a multivariate regression analysis. Furthermore, this study is monocentric, and even though it involved many surgeons who possessed at least ten years of experience, being a single-centre study limits its generalizability. 

## 6. Conclusions

A careful preoperative evaluation of a patient’s anatomic characteristics with a standardized CT scan assessment is crucial in a proper patient selection process. In this study, we identified an increased BSA (>1.9 m^2^), an annular diameter > 23 mm, a high calcium score (>2500 Agatson) and an increased aorta–sternum distance (>30 mm) aspredictors of longer CPB and X-clamp times (more than two-fold).

## Figures and Tables

**Figure 1 jcm-12-06717-f001:**
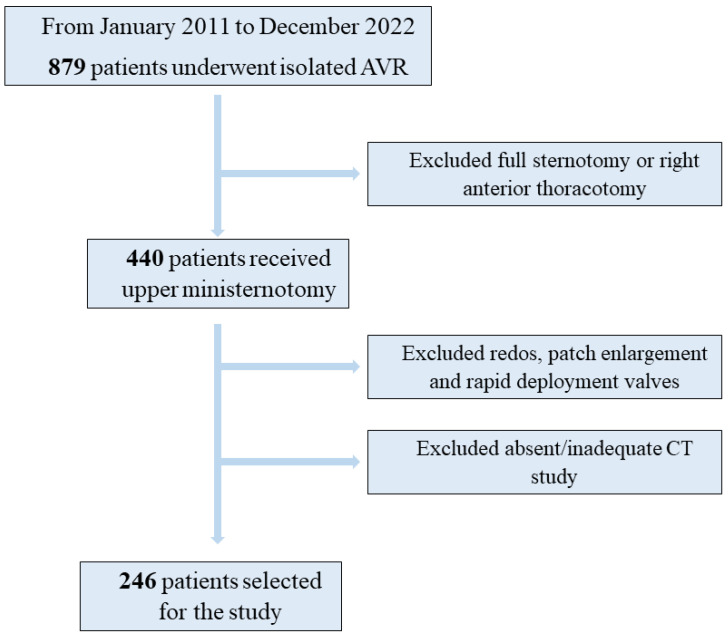
Flow chart showing the retrospective patient selection.

**Figure 2 jcm-12-06717-f002:**
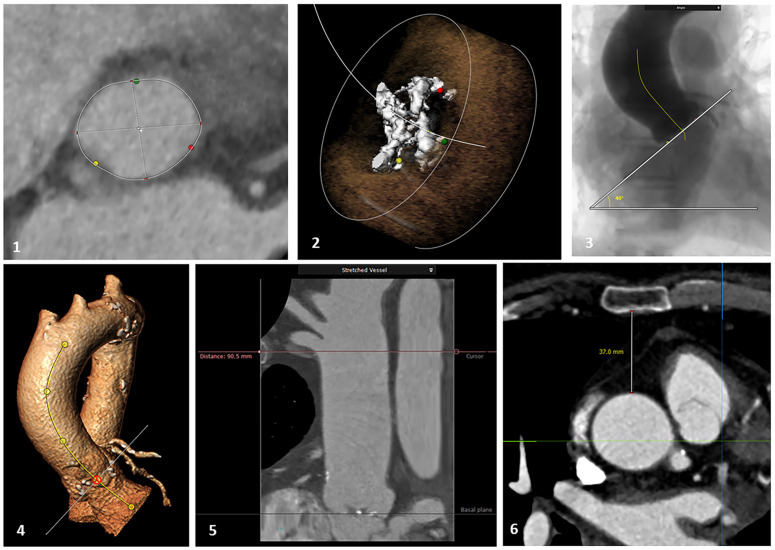
Example of the CT scan parameters assessed: (**1**) annular dimensions; (**2**) calcium score; (**3** and **4**) aortic annulus inclination angle; (**5**) ascending aorta length; (**6**) sternum/STJ distance.

**Figure 3 jcm-12-06717-f003:**
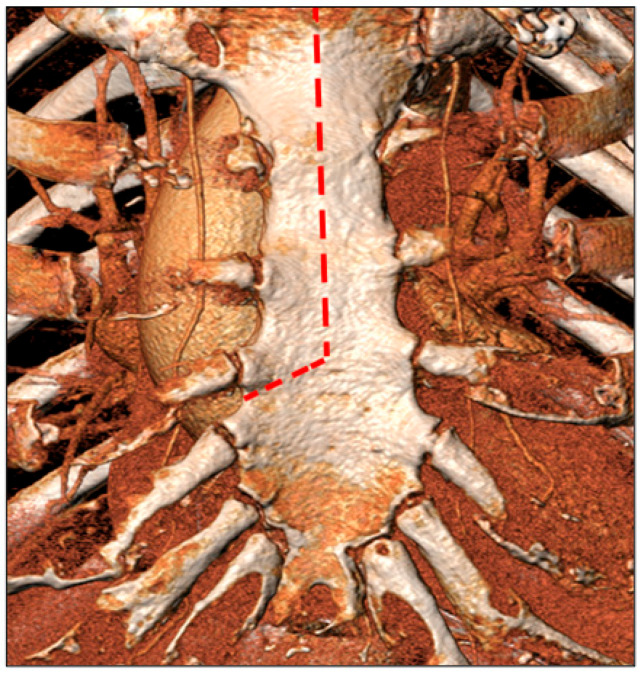
Schematic representation of J-ministernotomy at the 4th intercostal space using a 3D CT reconstruction.

**Figure 4 jcm-12-06717-f004:**
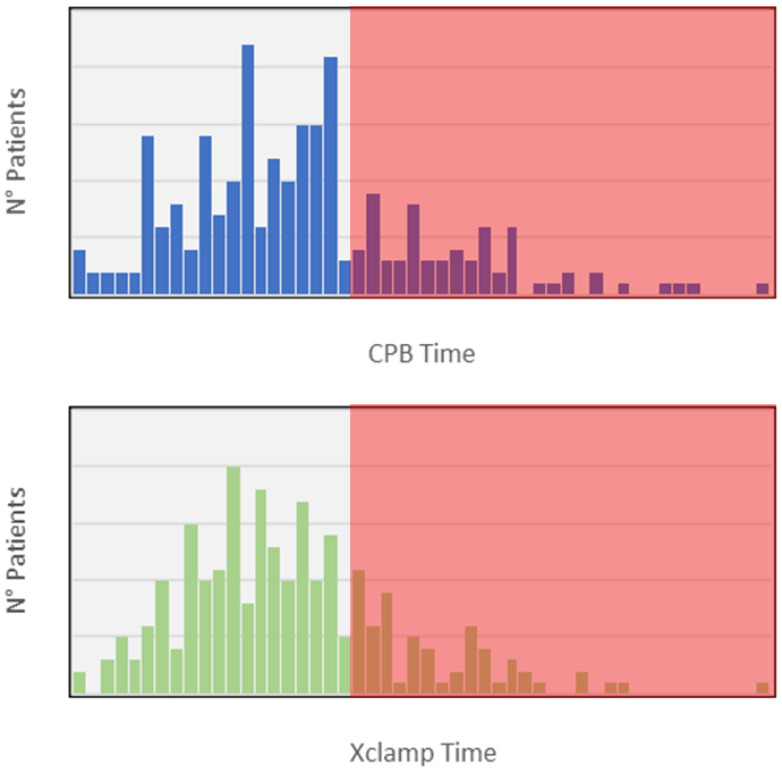
Diagrams showing the distributions of CPB and X-clamp times and the selection of patients are above the 75th percentile (red shadow).

**Figure 5 jcm-12-06717-f005:**
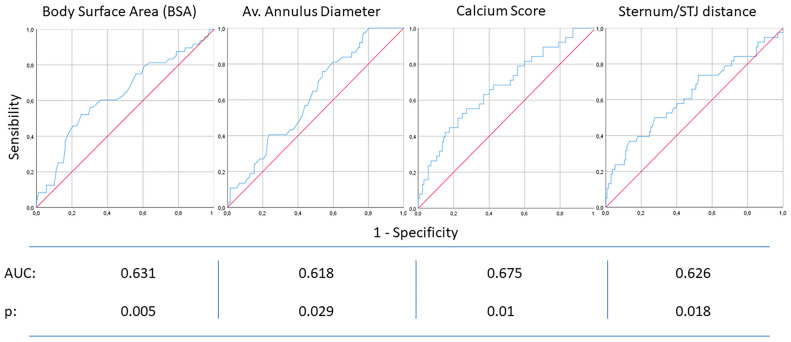
ROC curves for the continuous parameters associated with longer operative times (BSA, annulus diameter, calcium score, sternum/STJ distance); these ROC curves were used to choose the best cut-off points.

**Table 1 jcm-12-06717-t001:** Preoperative, intraoperative and postoperative patient features.

Preoperative Characteristics	
N° patients	246
Male sex	126 (51.2%)
Age, y	67 (12)
Height, cm	167 (9)
Weight, kg	74 (13)
Body surface area, m^2^	1.82 (0.19)
Body mass index, kg/m^2^	26.6 (3.8)
Prevalent aortic stenosis	198 (80.5%)
Prevalent aortic regurgitation	48 (19.5%)
Bicuspid aortic valve	62 (25.2%)
**Intraoperative data**	
Biological prosthesis	223 (90.7%)
Mechanical prosthesis	23 (9.3%)
Mean size prosthesis, mm	22.7 (2.0)
Arterial cannulation	
Ascending aorta	203 (82.5%)
Femoral artery	43 (17.5%)
Venous cannulation	
Right atrium	127 (51.6%)
Femoral vein	119 (48.4%)
CPB time, min	85.3 (20.3)
X-clamp time, min	64.0 (15.9)
**Postoperative outcomes**	
In-hospital death	5 (2.0%)
IABP	3 (1.2%)
ECMO	1 (0.4%)
Conversion to standard sternotomy	3 (1.2%)
Revision for bleeding	10 (4.0%)

CPB: cardiopulmonary bypass; IABP: intra-aortic balloon pump; ECMO: extra-corporeal membrane oxygenator.

**Table 2 jcm-12-06717-t002:** CT scan parameters.

Aortic annulus perimeter, mm	77.1 (8.9)
Aortic annulus average diameter, mm	24.0 (2.8)
Aortic valve calcium score, AU	2700 (2260)
Aortic annulus inclination, °	48.9 (11.0)
Ascending aorta length, mm	89.1 (12.0)
Sternum/STJ distance, mm	33.7 (10.0)

AU: Agatson units; STJ: sino-tubular junction.

**Table 3 jcm-12-06717-t003:** Univariate and multivariate logistic regressions, identifying predictive variables of longer procedures.

Univariate Regression	HR	95% CI	*p* Value
**Male sex**	**2.484**	**1.270–4.859**	**0.008**
Age	0.988	0.963–1.012	0.321
Height	1.025	0.989–1.063	0.174
**Weight**	**1.038**	**1.014–1.064**	**0.002**
**BSA**	**11.41**	**2.063–63.11**	**0.005**
**BMI**	**1.117**	**1.030–1.211**	**0.008**
Biological prosthesis	1.163	0.409–3.307	0.777
Prosthesis size	1.092	0.935–1.276	0.265
Central arterial cannulation	1.113	0.493–2.511	0.796
Central venous cannulation	1.452	0.769–2.739	0.250
Prevalent AR	2.543	0.714–9.055	0.150
BAV	0.495	0.190–1.293	0.151
**Annulus perimeter**	**1.059**	**1.016–1.103**	**0.007**
**Annulus diameter**	**1.164**	**1.021–1.327**	**0.023**
Ascending aorta length	1.015	0.983–1.047	0.366
**Calcium score (log)^+^**	**1.696**	**1.088–2.644**	**0.020**
**Sternum/STJ distance**	**1.052**	**1.014–1.092**	**0.007**
Aortic annulus inclination angle	0.988	0.955–1.021	0.464
**Multivariate regression**	**HR**	**95% CI**	***p*** **Value**
Male sex	1.218	0.432–3.434	0.709
BSA	5.946	0.424–83.48	0.205
Annulus diameter	1.010	0.832–1.227	0.917
Calcium score (log)^+^	1.532	0.954–2.462	0.078
Sternum/STJ distance	1.021	0.972–1.072	0.186

BSA: body surface area; BMI: body mass index; BAV: bicuspid aortic valve; TAV: tricuspid aortic valve; ^+^ for calcium score, logarithmic transformation was necessary to compare the data.

**Table 4 jcm-12-06717-t004:** Predictors of longer procedure according to selected cut-off values.

Predictors of a Longer Procedure	Odds Ratio	*p* Value
Male sex	2.5	<0.01
BSA > 1.9 m^2^	3.2	<0.01
Annulus diameter > 23 mm	2.9	<0.01
Calcium score > 2500 AU	2.9	<0.01
Sternum/STJ distance > 30 mm	2.2	0.02

**Table 5 jcm-12-06717-t005:** Comparison of selected variables (mean and frequency) among the two groups of short (below the 75th percentile) or long (above the 75th percentile) procedures.

Variables	Procedure Times < 75th Percentile	Procedure Times > 75th Percentile	*p* Value
N°	198 (80.5%)	48 (19.5%)	
CPB time, min	78.0 ± 13.7	114.9 ± 15.6	<0.001
X-clamp time, min	58.1 ± 9.8	88.3 ± 12.1	<0.001
BSA, m^2^	1.80 ± 0.18	1.89 ± 0.21	0.004
Annulus diameter, mm	23.7 ± 2.7	24.9 ± 2.7	0.019
Calcium score, Agatson units	2363 ± 1952	3868 ± 2798	<0.001
STJ/sternum distance, mm	32.6 ± 9.1	37.7 ± 12.2	0.005
Male sex, frequency	93 (46.9%)	33 (69.7%)	0.007

## Data Availability

The data presented in this study are available on request from the corresponding author.

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
