# Peer review of "Aortic Valve Replacement: Understanding Predictors for the Optimal Ministernotomy Approach"

_jcm, 2023, doi:10.3390/jcm12216717_

Round 1

Reviewer 1 Report

I reviewed with interest the manuscript of Francesco Giosuè Irace et al. "Aortic valve replacement: understanding predictors for the optimal ministernotomy approach". The article is devoted to the search for predictors of increased procedure duration in Aortic valve replacement with ministernotomy approach. The authors found specific indicators in the CT-Scan assessment associated with an increase in the duration of the operation by more than two times. These results may be useful in planning operations of this type. Nevertheless, when reviewing, I had questions and comments that I would like to receive answers from the authors.

1. The idea of using assessment CT scan for planning Aortic valve replacement operations with the ministernotomy approach is fairly well presented in the literature. In addition to the studies cited by the authors, the following can also be mentioned (1-4). I think it is advisable to add a consideration of the results of these studies to the text of the manuscript, for example, in the Discussion section.

2. In the Statistics section, it is not indicated which type of regression analysis the authors used (linear or logistic?).

3. In Table 3, significant statistical indicators for predictors of a longer procedure were found only with a single-variant logistic regression, with a multivariate such significant indicators were not identified. Why then do the authors believe that these measures were associated with longer surgery?

4. The title of table 3 is incomplete, it should be supplemented.

5. It is not clear why the authors claim that the factors they found are associated with an increase in the duration of the operation by more than two times. This is not clear from the results presented. Perhaps data should be given in two groups (in 48 patients (19.5%) with a “longer procedure” and other patients)?

6. The beginning of the Discussion section is more appropriate for the Introduction section. In addition, the authors repeatedly cite the purpose of the study in the discussion, this is redundant. It is advisable to start the Discussion section with the main results obtained by the authors.

References:

1.      Jug J, Štor Z, Geršak B. Anatomical circumstances and aortic cross-clamp time in minimally invasive aortic valve replacement. Interact Cardiovasc Thorac Surg. 2021 Jan 22;32(2):204-212. doi: 10.1093/icvts/ivaa251.

2.      Fabre O, Durand F, Hysi I. A Novel Computed Tomography Scan Tool for Patient Selection in Minithoracotomy Aortic Replacement. Ann Thorac Surg. 2020 Oct;110(4):e339-e341. doi: 10.1016/j.athoracsur.2020.03.111.

3.      Charchyan E, Breshenkov D, Belov Y. Preoperative assessment of the technical complexity of minimally invasive aortic root repair. J Cardiovasc Surg (Torino). 2023 Jun;64(3):322-330. doi: 10.23736/S0021-9509.22.12195-6.

4.      Li H, Castro M, Haigron P, Verhoye JP, Ruggieri VG. Decision support system for the planning of minimally invasive aortic valve replacement surgery. Int J Comput Assist Radiol Surg. 2018 Aug;13(8):1245-1255. doi: 10.1007/s11548-018-1725-7.

No comments

Author Response

I reviewed with interest the manuscript of Francesco Giosuè Irace et al. "Aortic valve replacement: understanding predictors for the optimal ministernotomy approach". The article is devoted to the search for predictors of increased procedure duration in Aortic valve replacement with ministernotomy approach. The authors found specific indicators in the CT-Scan assessment associated with an increase in the duration of the operation by more than two times. These results may be useful in planning operations of this type. Nevertheless, when reviewing, I had questions and comments that I would like to receive answers from the authors.

  1. The idea of using assessment CT scan for planning Aortic valve replacement operations with the ministernotomy approach is fairly well presented in the literature. In addition to the studies cited by the authors, the following can also be mentioned (1-4). I think it is advisable to add a consideration of the results of these studies to the text of the manuscript, for example, in the Discussion section.
  • Answer: Thank you for your precious suggestion. We modified and expanded the discussion section including the references you suggested
  • Change: Discussion section modified, LINES 230-237, References modified.
  1. In the Statistics section, it is not indicated which type of regression analysis the authors used (linear or logistic?).
  • Answer: Thanks for your comment. The analysis used was a logistic binary regression, as was performed after the categorization of outcome variable (procedure time). We will specify that in the methods section.
  • Change: Line 139, text modified.
  1. In Table 3, significant statistical indicators for predictors of a longer procedure were found only with a single-variant logistic regression, with a multivariate such significant indicators were not identified. Why then do the authors believe that these measures were associated with longer surgery?
  • Answer: Thank you for the opportunity to clarify this point. A major limitation of our study is its retrospective nature, for this reason many patients were not adequate to be included and the study results underpowered to demonstrate the validity of our anatomical predictors also in a multivariate analysis. We recognise this as an important limitation and will provide a dedicated paragraph as also suggested by Reviewer 2. Even if not conclusive, we hope that this study could open to further insights.
  • Change: Limitation section added

4. The title of table 3 is incomplete, it should be supplemented.

  • Answer: Thank you for the observation, we will correct the table title.
  • Change: Line 247, Table title modified

5. It is not clear why the authors claim that the factors they found are associated with an increase in the duration of the operation by more than two times. This is not clear from the results presented. Perhaps data should be given in two groups (in 48 patients (19.5%) with a “longer procedure” and other patients)?

  • Answer: Thank you for the opportunity for clarifying this point. Our assumption was not that operation times are increased by a 2-fold (or more) factor, but that the risk for a procedure of being “longer” is increased by two times. For the sake of clarity we added a new table (n 5), were the results regarding the selected variables are presented as comparison between groups, as you suggested.
  • Change: Line 255, Table 5 added

6. The beginning of the Discussion section is more appropriate for the Introduction section. In addition, the authors repeatedly cite the purpose of the study in the discussion, this is redundant. It is advisable to start the Discussion section with the main results obtained by the authors. 

  • Answer: Thank you for your suggestions. We will modify the manuscript accordingly.
  • Change: Introduction and Discussion section modified. LINES 100-116, 215-232

Reviewer 2 Report

The authors of this study presented data regarding predictors of optimal mini-sternotomy approach for aortic valve replacement.

Comments:

1. The authors state that surgical aortic valve replacement represents the treatment of choice for patients with severe aortic valve disease. This statement is incorrect because the decision regarding surgical versus trans-catheter aortic valve replacement is far more nuanced. Based on randomized clinical trial data, trans-catheter aortic valve replacement has become the preferred approach for aortic valve replacement in majority of patients with severe senile/calcific aortic stenosis. Please elaborate further on patient and anatomic factors that guide the choice between surgical versus trans-catheter aortic valve replacement.

2. Did any of the patients in this cohort have a pre-existing aortic valve prosthesis?

3. How many patients had undergone prior cardiac surgery or sternotomy?

4. It would be helpful if the authors could share data regarding pre-operative predictors for patients who required conversion to standard sternotomy.

Minor grammatical revision required. 

Author Response

. The authors state that surgical aortic valve replacement represents the treatment of choice for patients with severe aortic valve disease. This statement is incorrect because the decision regarding surgical versus trans-catheter aortic valve replacement is far more nuanced. Based on randomized clinical trial data, trans-catheter aortic valve replacement has become the preferred approach for aortic valve replacement in majority of patients with severe senile/calcific aortic stenosis. Please elaborate further on patient and anatomic factors that guide the choice between surgical versus trans-catheter aortic valve replacement.

Answer: We thank the reviewer to give us the opportunity to clarify that point, we modified the sentence. 

Change: LINE 66 modified

  1. Did any of the patients in this cohort have a pre-existing aortic valve prosthesis?

Answer: No, reoperations were excluded (LINE 126)

Change: None

3. How many patients had undergone prior cardiac surgery or sternotomy?

Answer:  No, reoperations were excluded

Change: none

4. It would be helpful if the authors could share data regarding pre-operative predictors for patients who required conversion to standard sternotomy

Answer: the three conversions to full sternotomy occurred due to excessive bleeding after weaning from CPB (Line 180); for this reason we did not perform any analysis for predictors (being this analysis beyond the purposes of our study)

Reviewer 3 Report

Abstract:

- Sums up the main points sufficiently.

- line 14: After "bypass", the word "time" is crossed out and should bei left out

- Methods and results are missing the period after the last sentence.

Background: 

- line 41-42: "may underwent" should read "may undergo".

- line 55: Instead of "...is rising" it should probably say "...has been pioneered." as these are the first cases of robotic aortic surgery in humans.

Patients and Methods:

- As this is a retrospective study, it is not clear why all 246 patients included in the study were examined with preoperative ECG-gated CT. As discussed in the chapter before, this is not standard for SAVR. Where the patients evaluated for both TAVR und SAVR or were there other indications for ECG-gated CT imaging? Was CT imaging used to choose patients for M-SAVR? Please elaborate!

- Even though this is not a radiological journal, basic information about the technical parameters of CT imaging (type and slice count of CT scanner, type of ECG-Gating, flow rate and volume of contrast agent) should be included

Disscussion

- It should be discussed why aortic valve inclination was not correlated to procedure time, which contrasts citation (10) where the access angle was a predictor of procedure complexity

- A paragraph about the limitation of this study should be included. In particular, this study was not adequately powered to assesss the impact of CT parameters on patient outcome.

There are some minor grammatical errors.

Author Response

Background:

- line 41-42: "may underwent" should read "may undergo".

- line 55: Instead of "...is rising" it should probably say "...has been pioneered." as these are the first cases of robotic aortic surgery in humans.

  • Answer: Thank you for your suggestion, we will modify the text accordingly
  • Change: LINE 73, 85 text modified

Patients and Methods:

- As this is a retrospective study, it is not clear why all 246 patients included in the study were examined with preoperative ECG-gated CT. As discussed in the chapter before, this is not standard for SAVR. Where the patients evaluated for both TAVR und SAVR or were there other indications for ECG-gated CT imaging? Was CT imaging used to choose patients for M-SAVR? Please elaborate!

  • Answer: Than you to give us the opportunity to clarify this point: most of patients treated at our institution receive a cardiac CT scan in the preoperative work up, both for coronary study, avoiding the need of coronary angiography when possible, and for anatomical assessment (aortic position, aortic wall calcification). We added this specification in the methods section
  • Change: LINE 116-9, methods section modified

- Even though this is not a radiological journal, basic information about the technical parameters of CT imaging (type and slice count of CT scanner, type of ECG-Gating, flow rate and volume of contrast agent) should be included

  • Answer: Than you to give us the opportunity to clarify this point. We added the CT specifics in the methods section.
  • Change: LINE 119-124 methods section modified.

Disscussion

- It should be discussed why aortic valve inclination was not correlated to procedure time, which contrasts citation (10) where the access angle was a predictor of procedure complexity

  • Answer: Thank you for the opportunity to clarify this. Elattar et al. considered the angle between aorta and access plane, while we considered the inclination angle of the aortic annulus on a horizontal plane (respect to the floor)
  • Change: Discussions section modifed, LINE 245-7

- A paragraph about the limitation of this study should be included. In particular, this study was not adequately powered to assesss the impact of CT parameters on patient outcome.

  • Answer: Thank you for you observations, we added a limitation section to address your concerns.
  • Change: LINE 230, Limitation section added.

Round 2

Reviewer 1 Report

The authors responded to my comments and made corrections to the text of the manuscript. I have no other comments.

No comments